# Amyloidosis and Glomerular Diseases in Familial Mediterranean Fever

**DOI:** 10.3390/medicina57101049

**Published:** 2021-10-01

**Authors:** Rossella Siligato, Guido Gembillo, Vincenzo Calabrese, Giovanni Conti, Domenico Santoro

**Affiliations:** 1Unit of Nephrology and Dialysis, Department of Clinical and Experimental Medicine, University of Messina, 98125 Messina, Italy; guidogembillo@live.it (G.G.); v.calabrese@outlook.it (V.C.); santisi@hotmail.com (D.S.); 2Department of Biomedical, Dental, Morphological and Functional Imaging Sciences, University of Messina, 98125 Messina, Italy; 3Pediatric Nephrology Unit, AOU Policlinic “G Martino”, University of Messina, 98125 Messina, Italy; giovanniconti@hotmail.com

**Keywords:** Familial Mediterranean fever, serum amyloid A, AA amyloidosis, glomerulonephritis, vasculitis, nephrotic syndrome, NGAL, anakinra, canakinumab, tocilizumab

## Abstract

Familial Mediterranean fever (FMF) is a genetic autoinflammatory disease with autosomal recessive transmission, characterized by periodic fever attacks with self-limited serositis. Secondary amyloidosis due to amyloid A renal deposition represents the most fearsome complication in up to 8.6% of patients. Amyloidosis A typically reveals a nephrotic syndrome with a rapid progression to end-stage kidney disease still. It may also involve the cardiovascular system, the gastrointestinal tract and the central nervous system. Other glomerulonephritis may equally affect FMF patients, including vasculitis such as IgA vasculitis and polyarteritis nodosa. A differential diagnosis among different primary and secondary causes of nephrotic syndrome is mandatory to determine the right therapeutic choice for the patients. Early detection of microalbuminuria is the first signal of kidney impairment in FMF, but new markers such as Neutrophil Gelatinase-Associated Lipocalin (NGAL) may radically change renal outcomes. Serum amyloid A protein (SAA) is currently considered a reliable indicator of subclinical inflammation and compliance to therapy. According to new evidence, SAA may also have an active pathogenic role in the regulation of NALP3 inflammasome activity as well as being a predictor of the clinical course of AA amyloidosis. Beyond colchicine, new monoclonal antibodies such as IL-1 inhibitors anakinra and canakinumab, and anti-IL-6 tocilizumab may represent a key in optimizing FMF treatment and prevention or control of AA amyloidosis.

## 1. Introduction

Familial Mediterranean fever (FMF) is an autosomal recessive autoinflammatory disease, characterized by periodic rapid fever attacks associated with self-limited serositis, like pericarditis and peritonitis, with prodromal symptoms [1]. The onset of FMF is usually comprised between 5 and 15 years old, with 90% of patients developing the disease before 20 years-old. Patients are usually asymptomatic between the attacks, which may happen once a year or up to twice a day [1].

Other features are monoarticular arthritis affecting typically the lower limbs and erysipelas-like skin lesions [2,3]. Fever attacks are associated with an increase of acute-phase proteins, especially C-reactive protein (CRP) and serum amyloid A (SAA), and a rise of inflammatory markers such as *erythrocyte sedimentation rate* (ESR).

FMF is more frequent in Mediterranean populations as Turks (with a prevalence of 1:400–1:1000), Armenians (1:500), Jews and Arabs. The responsible of FMF is the mutated pyrin/marenostrin, a protein composed of 781 amino acids that can regulate interleukin 1β (IL-1β) production. Pyrin interacts with caspase-1 and the nucleotide-binding domain leucine-rich repeat/pyrin domain-containing-3 (NLRP3) [4], which are both fundamental parts of the inflammasome complex. Most pyrin mutations are located in the C-terminal domain which binds procaspase 1 and reduces IL-1β activation. In 1997, two different consortia described for the first time the *MEFV* gene site in chromosome 16p13.3, consisting of 10 exons [5,6]. Almost 317 variants of this gene are known, but only nine are pathogenic such as Met694Val, Met694Ile, Ala744Ser, Val726Ala, Met680Ile, Met694Ile, Pro369Ser, Glu148Gln. Aktas and colleagues analyzed 259 Turkish patients and found homozygous *MEFV* mutations in 31.9%, while heterozygous mutations were 35.6% and compound heterozygous mutations 27.5%. Higher severity score was associated with the homozygous group (50.6%, *p* < 0.0001), while it was less frequent in the heterozygous (13.6%) and the compound heterozygous groups (14.7%). The only significant difference in the clinical presentation was found for erysipelas-like erythema which had a greater incidence in the homozygous (69.6%) than in heterozygous (37.5%, *p* < 0.0001) [7]. An Italian cohort of patients and their first-degree relatives reported a higher prevalence of heterozygous genotype (up to 85.98%). They demonstrated no significant difference in clinical phenotype except for p.Met694Val/wt and p.Met680Ile/wt genotype, which are associated with the most severe clinical features [8].

FMF diagnosis according to Tel Hashomer criteria requires the presence of at least one among major criteria:Recurrent fever attacks with serositis;AA amyloidosis, without other predisposing diseases;Symptoms responsive to colchicine;

And two of minor criteria:
Recurrent fever episodes;The presence of erysipelas-like erythema;Family history of FMF;

According to the patients’ phenotype, we can identify three main patterns of FMF [9]:(1)Type 1, with short episodes of inflammation and serositis;(2)Type 2, characterized by the onset of secondary amyloidosis without any other overt systemic symptoms of FMF;(3)Type 3 noted as “silent homozygous or compound heterozygote state” without relevant symptoms.

Varan et al. showed how, in particular, homozygous mutation M694V results significantly more common in FMF patients with chronic inflammation (*p* < 0.001), who are at higher risk of developing secondary amyloidosis [10].

The role of vitamin D-binding protein (VDBP) has been recently investigated in FMF. VDBP is a 458 amino acid protein with two domains binding vitamin D and actin respectively, located in correspondence of 35–49 and 373–403 residues. A trial on 107 FMF patients (52 without any *MEFV* mutation) and 25 healthy individuals, analyzed the *VDBP* rs4588 and rs7041 polymorphisms, resulting in the identifications of six genotypes. Allelic variant 2 was significantly more frequent in the subgroup of FMF patients *MEFV(-)* than in healthy controls (*p* = 0.001) and the carriers also had a higher risk of development of amyloidosis or arthritis (*p* = 0.026) than other FMF patients into the same group [11]. This effect may be a consequence of the dysregulation of vitamin D levels and the impairment of its immunomodulatory and anti-inflammatory actions. Vitamin D may counteract the activation of the NF-κB pathway both via VDR-mediated sequestration of NF-κB signaling products and by inhibiting NF-κB transactivation through the modulation of advanced glycation end-products and their receptor (AGE-RAGE system) [12,13]

Both heterozygous and homozygous deletion polymorphisms of the *angiotensin-converting enzyme* (ACE) gene, located on chromosome 17q23, may be associated with a higher risk of FMF. Angiotensin II plays a role in the regulation of inflammatory cells recruitment, interacting with interleukin-4 (IL-4), IL-6, tumor necrosis factor-α (TNF-α) and monocyte chemoattractant protein (MCP). D/D genotype was linked to an increased risk of developing FMF [*p* < 0.001; OR (95%): 7.715 (4.503–13.22)] and, in particular, D/D+I/D genotype and fever were significantly associated (*p* = 0.04) [14]. In the same trial, *IL-4* gene P1/P1 genotype was also related to FMF diagnosis (*p* < 0.001), confirming its direct pathogenic role in that setting [15]. Nursal and colleagues confirmed that *ACE* I/D and D/D variants appear significantly more frequent in FMF patients than in a healthy control group (*p* < 0.05). In addition, particular I/D+I/I genotypes were associated with the development of amyloidosis in a subgroup analysis of FMF patients with and without this complication (*p*
*<* 0.03, OR 3.24; 95% CI 1.05–12.01) [15].

A study cohort of 374 patients diagnosed with secondary amyloidosis followed at the U.K. National Amyloidosis Centre for 15 years, recognized FMF as the primary disorder in 5% of cases [16]. Nevertheless, in those populations in which MEFV mutation is more frequent, FMF may represent one of the leading causes. Secondary amyloidosis is the most common and severe renal complication because it may evolve in end-stage renal disease (ESRD) in 5–10 years from the onset of proteinuria, especially if patients are untreated, scarcely compliant, or non-responsive to colchicine [17]. Other renal diseases develop in 22% of cases in FMF and include recurrent pyelonephritis, urinary abnormalities such as hematuria and/or proteinuria, glomerular involvement with IgA nephropathy, membranous nephropathy, mesangioproliferative or rapid progressive glomerulonephritis, and vasculitis [18,19].

## 2. A Focus on Secondary Amyloidosis

Amyloidosis was first addressed by Virchow in 1854 as extracellular deposition of fibrillary substances resembling cellulose when observed at light microscopy. Irrespective of the specific native protein, all amyloid deposits are characterized by insoluble polypeptides misfolded to a common β-sheet conformation with interstrand distances of 4.7 Å and intersheet distances of 10–13 Å. Polypeptides fold in β-sheet chains constitute a protofilament, while several of them twisted around one another form an amyloid fibril, with a 7.5–10 nm diameter visible on transmission electron microscopy [20]. Amyloid deposits can be identified by Congo red staining under a polarized light microscope, which confers a typical apple-green birefringence.

Amyloidogenesis may be due to specific genetic mutations that confer intrinsic instability of the primary amino acid structure, as in hereditary amyloidosis. Conditions such as increased concentrations of the precursor protein and/or altered extracellular environment (presence of oxidative species or metallic ions, modified local pH or temperature) may lead to ineffective proteolysis and generation of insoluble fragments in acquired forms of amyloidosis [21]. More protofilaments can bind together in the presence of other proteins acting as a scaffold, such as heparan sulfate proteoglycan (HSPG) or serum amyloid P-component (SAP), which have ubiquitary distribution in different classes of amyloidosis. New evidences also show an active role of apolipoproteins apo-AI, apo-AIV, and apo-E [22]. Renal amyloidosis, in particular, may be favoured by glomerular basement membrane (GBM) composition rich in glycosaminoglycans, which do not offer only support for progressive nucleation of amyloid fibrils, but may also actively impair their degradation [23].

Up to now, 22 different forms of localized amyloidosis have been identified and involve mainly the central nervous system, respiratory tract, bladder, skin and cornea. Systemic amyloidosis types, irrespective of their pathogenesis, usually share renal, cardiovascular and gastrointestinal involvement. Globally, amyloidosis has been distinguished in:hereditary forms, due to genetic variants of amyloid precursors such as transthyretin (TTR), fibrinogen Aα, ApoAI, ApoAII, ApoAIV, ApoCII, ApoCIII, lysozyme, gelsolin, β2-microglobulin (β2M) and cystatin;primary forms due to acquired plasma cells disorders with over-expression of monoclonal immunoglobulin (Ig) light chains (AL), rarely of Ig heavy chains (AH) or both (AHL);secondary amyloidosis (AA) due to chronic inflammatory processes which enhance the synthesis of acute-phase reactants such as SAA or CRP production and progressive deposition;senile amyloidosis, due to misfolding wild-type TTR (ATTRwt);Aβ2M wild-type, especially in patients requiring renal replacement therapy (RRT) with hemodialysis.

Renal involvement in amyloidosis is represented mainly by capillary and mesangial deposition of amorphous material at light microscopy, which may also assume a nodular aspect when in considerable amount. Still, it appears weakly stained by periodic acid-Schiff (PAS) compared to diabetic nephropathy nodules of Kimmestiel-Wilson that are mainly composed of glycosaminoglycans. Advanced amyloidosis deposits may also be located in tubule-interstitium and lead to chronic damage features of tubular atrophy and interstitial fibrosis.

Differential diagnosis between the different types of amyloidosis may be provided by immunofluorescence on biopsy samples with no detection of complement, fibrinogen, immunoglobulin or light chains in AA amyloidosis. In this case, only anti-SAA protein antibodies may be positive and localized in correspondence of deposits, representing a more reliable indicator than the classic practice of treating the sample with potassium permanganate to demonstrate the loss of Congo red staining, typical of AA amyloidosis.

An attempt to encode a renal amyloid prognostic score (RAPS) took into account the glomerular, vascular and interstitial amyloid A deposition, as well as glomerular sclerosis, interstitial inflammatory infiltration and/or fibrosis and tubular atrophy. Although renal survival was significantly lower among patients with RAPS grade III, at further analysis, RAPS grade and glomerular amyloid deposition resulted to be associated with baseline eGFR level and proteinuria respectively. In contrast, only extensive glomerular amyloid deposition turned out to be an independent predictor for the development of ESRD [24].

Nephrotic syndrome is a typical presentation of renal amyloidosis, with proteinuria (mainly albuminuria), and hypoalbuminemia, associated with diuretic-refractory edema. In addition, the involvement of cardiovascular and of autonomic nervous systems can cause hemodynamic instability with additional limitations to the use of diuretics. On the contrary, when amyloid has a main interstitial or vascular deposition, the clinical presentation may differ with minimal proteinuria but a sensible reduction in GFR. Renal impairment tends to progress less rapidly when tubulointerstitial deposition predominates rather than glomerular.

Kukuy et al., recently described a different clinical evolution of AA amyloidosis, which was defined as “amyloid storm” to indicate a rapid progression of serum creatinine and proteinuria in two weeks, usually triggered by infections. The authors evaluated retrospectively 40 FMF patients’ and found a positive correlation with the primary endpoint of ESRD and/or death within a year in people who had at least an episode (16 patients in the study arm vs. 3 in the control group with more stable renal function) [25]. Patients who never experienced this rapid renal impairment had typically a slower rise of serum creatinine to 1.2 mg/dl (41.55 ± 10.98 years since FMF onset vs. 26.5 ± 15.15 years *p* = 0.001), culminating in older age at study entry (48.9 ± 9.98 years vs. 39.95 ±16.81 respectively, *p* = 0.05) [26].

## 3. The Role of SAA in FMF and AA Amyloidosis Follow-Up

Inflammatory diseases such as periodic fever syndromes, rheumatoid arthritis, inflammatory bowel diseases, cystic fibrosis as well as chronic infections, may induce AA amyloidosis due mainly to deposition of 76 residues of N-terminal fragment of SAA, and rarely to fragments of different lengths [26,27,28].

Within *SAA* genes, only *SAA1* and *SAA2* encode two inducible isotypes of the acute phase serum protein, with several allelic variants (*a, b, g* in *SAA1* and *a, b* in *SAA2*). In particular, single nucleotide polymorphisms (SNPs) within the exon 3 of the *SAA1* may represent risk factors for the development of AA amyloidosis, as demonstrated for isoform SAA1a (52 Valine/57 Alanine) in Caucasian and SAA1g (52 Alanine/57 Valine) in Japanese population [27].

SAA protein exerts a cytokine-like role, enhancing the expression of pro-IL-1β, IL-6, cyclooxygenase-2 (COX-2) and prostaglandin E 2 (PGE2), as a part of inflammatory processes. These effects of SAA1 are mediated in several organs through activation of the NF-κB, p38 and ERK1/2 pathways via either the toll-like receptor 2 (TLR2) and 4 (TLR4), either scavenger receptor class B type 1 (SRB1), formyl peptide receptor-like 1 (FPRL1), or RAGE [27,29,30]. Moreover, SAA may directly activate the NLRP3 inflammasome and subsequently caspase-1, with a possible pathogenic role in autoinflammatory diseases, more than being only a marker of diseases’ activity. Conversely, TNF-α, IL-1β and IL-6 may exert a positive feedback on SAA synthesis, in a self-maintenance process.

The periodic evaluation of SAA titer has a central role in the clinical management of FMF patients, because of its higher sensitivity in detecting the subclinical residual inflammation even in attack-free periods compared to CRP or ESR [31].

Although it cannot be adopted as a predictive factor for the onset of secondary amyloidosis, SAA demonstrated reliability in the follow-up of patients who have already developed this complication. In particular, Lachmann and colleagues first showed that a median annual SAA concentration ≥ 155 mg/L in AA amyloidosis patients conferred a relative risk of death of 17.7 times higher than SAA titer < 4 mg/L in patients with more effective control of inflammation (95% CI 8.4–36.0, *p* > 0.001). In this observational study, 5% of the sample included patients affected by FMF undergoing colchicine treatment. In the subgroup of AA amyloidosis patients with glomerular filtration rate (GFR) > 20 mL/min at baseline, renal survival was significantly impaired if median SAA levels reached 28 mg/L (*p* < 0.001, by the Kruskal–Wallis test). Moreover, amyloid deposits regressed in up to 60% of patients who had a median SAA concentration <10 mg/L, with a better survival rate (*p* = 0.04) [18].

SAA is also a consistent predictive marker of atherosclerosis in FMF, directly correlated to the intima-media thickness of the common carotid artery (CIMT) [32] and the development of vasculopathy. In particular, Sargsyan et al. demonstrated that SAA levels were correlated to increased endothelin-1 (ET-1) titers (*r* = 0.459, *p* = 0001) in 30 FMF patients affected by coronary heart disease, pulmonary hypertension, stroke, vasculitis (polyarteritis nodosa, IgA vasculitis, livedoid vasculopathy) and Raynaud’s phenomenon, compared to non-vasculopathic FMF patients and healthy controls, concluding that SAA may also represent a surrogate marker of endothelial injury [33].

## 4. Glomerular Diseases and FMF

Since 1989, cases of other glomerulopathies than amyloidosis have been reported in FMF patients. Several associations between FMF regard IgA nephropathy (IgAN) [34,35,36,37], membranous nephropathy (MN) [36,38], membranoproliferative glomerulonephritis (MPGN) [36,39,40], mesangioproliferative glomerulonephritis (MsPGN) [36,41,42,43], focal and segmental glomerulosclerosis (FSGS) [36,37], vasculitis [44,45,46,47]. Rarer cases are represented by and rapid progressive glomerulonephritis (RPGN) [48], IgM nephropathy [49,50] and fibrillary glomerulopathy [51].

A retrospective study involving 64 adult patients (18 to 64 years-old) with primary glomerulonephritis and FMF symptoms, detected *MEFV* mutations in 35.9% of the cohort, of which 30% was affected by MsPGN, 30% by MN, 17% by IgAN, 8.6% immune complex glomerulopathy, 8.6% FSGS, 4.3% minimal change disease. Interestingly, the heterozygous E148Q genotype was prevalent in 10.9% of *MEFV*(+) [36]. Of these patients, three were IgAN, two MsPGN, one MN, and one FSGS. Two clinical reports regarding mesangioproliferative glomerulonephritis patients confirm the association with E148Q genotype [43,44], but it does not appear to be disease-specific as another case of MsPGN reported by Cagdas happened in a carrier of a different M694V/M694V homozygous mutation and responded to colchicine [42].

A similar preponderance in E148Q genotype was obtained by Gershoni-Baruch in an observational study on IgA vasculitis (IgAV) in FMF patients, pointing out the presence of compound heterozygous M694V/E148Q or homozygous V726A/V726A mutations in 12 out of the 52 IgAV patients evaluated [44]. This evidence was reaffirmed in another case report by Sozeri et al., with the detection of the same M694V/E148Q mutation in an adolescent affected by IgA vasculitis successfully treated with methylprednisolone pulses followed by oral prednisone and cyclophosphamide [45].Among vasculitis, IgAV is the most common vasculitis in FMF patients with a prevalence of 2.7–7%, followed by polyarteritis (PAN) with 0.9–1.4%, and Behçet’s disease (BD). A peculiar feature of IgA vasculitis in these patients is the higher incidence of intussusception (8.7%) as intestinal complication and a paradoxical lower amount of IgA deposits on biopsy samples compared to patients without FMF. PAN overimposed in FMF patients has an earlier onset (at a mean age of 17.9 years), and tends to present more perirenal hematomas (49%), glomerular and central nervous system involvement (33% and 31%, respectively) compared to PAN alone. The majority of patients affected by BD were described in the literature with worse dermatological, gastrointestinal, and neurological symptoms [52].

Up to now, the allele frequency of the three most common *MEFV* mutations E148Q, V726A, and M694V in IgA nephropathy cases is comparable to the ethnically adjusted general population distribution, and does not demonstrate a disease specificity nor a correlation with the clinical course of this glomerulonephritis in FMF patients [32,33,34,35].

Only two cases of IgM Nephropathy have been reported in association with FMF. In particular, Peru and colleagues described a *MEFV* M694V/M694 V mutation in exon 10 in a 9-years-old child. The same mutation was also found in two patients affected by MPGN C3+ and IgM+, both responsive to corticosteroids [50,51].

Rapid progressive glomerulonephritis has a lower incidence, with only two cases reported by Said et al. [48], while a unique case of fibrillary GN has been described in 2003, with the detection of the *MEFV* heterozygous missense mutation M680I [52].

## 5. The Nephrologist’s Point of View

FMF patients, according to the age of onset of their symptoms, are usually referred to a pediatrician or rheumatologist. However, in consideration of the complexity of clinical manifestations, patients need a multidisciplinary management of their disease. Moreover, the vagueness of early symptoms may impair a rapid diagnosis thus exposing patients to the risk of arising of possible complications.

Progressive kidney impairment is still a crucial point in the natural course of FMF and the nephrologist has to be actively involved in patients’ treatment. In particular, the follow-up must include a quarterly or biannual evaluation of complete blood count, CRP or SAA, liver enzymes, serum creatinine and urinalysis to monitor both disease activity and eventual colchicine toxicity [53]. SAA is the best choice to evaluate inflammatory status in attack-free intervals, but the assay may not be available in all the laboratories, so the clinicians may be forced to rely mostly on CRP in remote settings.

Urinalysis has a pivotal role as an easy and effective test to detect the possible onset of proteinuria. In case of a significant amount of microalbuminuria/urinary creatinine ratio [54], a complete kidney function evaluation must be carried out through determination of GFR, 24h proteinuria, *blood urea* nitrogen (BUN), electrolytes, serum protein electrophoresis and eventually serum and urinary immunofixation if required in the differential diagnosis between other causes of amyloidosis, such as AL or AH forms. The urinary sediment examination with phase-contrast microscopy may also be useful to detect glomerular microhematuria (with dysmorphic erythrocytes and/or acanthocytes) which could also help diagnose another glomerulonephritis that may equally affect FMF patients.

Eventually, kidney biopsy is still the gold standard exam for the differential diagnosis of the specific glomerulopathy and the quantification of the amyloid renal burden, and the amount of chronic kidney damage, to guide the clinicians to the most appropriate therapeutic choice.

Colchicine represented a real game-changer in the treatment of FMF since 1972, with a reduction in the incidence of AA amyloidosis from around 50% to 8.6% of patients [55,56]. The anti-inflammatory properties of colchicine are due to its regulation of neutrophils chemotaxis and activation, and inhibition of the release of superoxide and IL-1, IL-8 via the NALP3-inflammasome pathway. Moreover, colchicine also shows endothelial-directed activities, impairing neutrophils adhesion and inhibiting vascular endothelial growth factor (VEGF) stimulation [57].

The treatment of overlapping glomerulonephritis has been described in several case reports. In Huzmeli et al. experience, the treatment choice has to be evaluated according to GFR, proteinuria and specific histopathological lesions, as well as the FMF phenotype of the patients. FMF symptomatic patients (type 1) without nephrotic range proteinuria, were treated with colchicine and angiotensin receptor blockers (ARB), reaching 75% of complete remission. In four cases with phenotype 2 and non-nephrotic proteinuria, the treatment of choice was exclusively ARB, with 50% of full recoveries and a stability in proteinuria levels or only partial remission in other two patients, respectively. Phenotype 3 FMF with subclinical inflammation did not receive colchicine as well but ARB and/or immunosuppressive treatment according to the primary glomerulonephritis, with a lower complete remission rate.

## 6. Future Perspectives: Biomarkers and New Therapeutic Strategies

Neutrophil gelatinase-associated lipocalin (NGAL) has been recently investigated as a possible early indicator of nephropathy in FMF patients. NGAL is an extracellular protein usually filtered by glomeruli and reabsorbed via endocytosis on the tubular brush border, representing a sensible urinary marker of acute renal damage [58]. Oksay and colleagues evaluated serum and urinary markers in a cohort of 45 attack-free FMF children and 38 healthy controls of comparable ages, demonstrating significantly higher urinary NGAL (*p* = 0.0001) and urinary NGAL/urinary creatinine ratio (uNGAL/uCr) (*p* = 0.011) in the patients’ group than in controls. Moreover, both uNGAL level and uNGAL/uCr ratio were positively correlated to the number of attacks/year in FMF children (*r* = 0.743, *p* = 0.001 and *r* = 0.516, *p* = 0.001; respectively) [59].

New perspectives on therapies come from trials on biological agents targeting IL-1 (anakinra, canakinumab) and IL-6 (tocilizumab) [58,60].

Anakinra, a new recombinant homologue of IL-1 receptor antagonist (IL-1Ra) demonstrated its efficacy in colchicine-resistant FMF patients both in randomized clinical trials and observational studies. In long-term follow-up trials, the complete remission was achieved in 12–100% of subjects [61,62,63,64,65]^.^ Patients with FMF-related amyloidosis experienced the improvement or stabilization of renal function and proteinuria after the administration of anakinra [61,66,67,68,69].

Canakinumab, a human monoclonal anti-IL-1β antibody, has been adopted in small cohorts of patients [68,70,71]. A randomized trial including patients affected by colchicine-resistant FMF demonstrated a complete response in 61% of the treatment group vs. 6% of placebo group at 16 weeks (*p* < 0.001), with 46% of patients maintaining remission with canakinumab longer interval dose of 8 weeks [72]. Yazilitas and colleagues also showed as a secondary outcome the increase of GFR and stable proteinuria after 3 months in 2 of 3 of the patients affected by AA amyloidosis of 11 total FMF patients included in their cohort [73].

In a retrospective observational trial, seventeen FMF patients with AA amyloidosis in treatment with colchicine at maximum-tolerated dose were initially administered anakinra. Five patients switched to canakinumab for adverse reactions (leukopenia or injection site reactions) and two for inefficacy. Inflammatory markers such as CRP and ESR significantly reduced and reached normal levels in twelve patients. Except for six patients of the cohort who were undergoing renal replacement therapy, proteinuria improved in all the other patients after the association of IL-1 inhibitors to colchicine (median 1606 mg/day vs. 519 mg/day, *p* = 0.008) during a median of 16-months follow-up [74].

Tocilizumab is another human monoclonal anti-IL-6 antibody adopted in rheumatoid arthritis and juvenile idiopathic arthritis. It has been tested in three trials with promising results both in terms of attacks control and renal outcomes, with stable GFR and reduction of proteinuria [75,76,77].

## 7. Conclusions

AA amyloidosis has usually a clinical presentation characterized by nephrotic syndrome. Patients affected by autoimmune or autoinflammatory diseases such as FMF, should be routinely evaluated with urinalysis for early detection of proteinuria, given their risk of developing secondary amyloidosis. In particular, the interval between FMF diagnosis and AA amyloidosis onset may be long but, once recognized, CKD progression to ESRD is often rapid. For this reason, nephrologists have to be actively involved in the clinical follow-up of patients.

The adoption of more sensitive and specific markers is desirable to implement therapeutic strategies to prevent complications or, at least, to slow down their progression. SAA protein turned out to be a reliable surrogate marker of AA amyloidosis patients and renal survival, also reflecting the stability or regression of amyloid deposits. In this setting, urinary NGAL may be a precocious marker to detect the earliest signal of kidney impairment before microalbuminuria/uCr ratio or the onset of clinically significant proteinuria.

Finally, the adoption of new therapies such as IL-1 inhibitors and anti-IL-6 may represent a useful tool, but larger clinical trials are required to confirm their efficacy in those patients non-responder or intolerant to colchicine, which still remains a cornerstone in FMF clinical management.

## Data Availability

Not applicable.

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
