# Peer review of "Amyloidosis and Glomerular Diseases in Familial Mediterranean Fever"

_medicina, 2021, doi:10.3390/medicina57101049_

Round 1
Reviewer 1 Report
Overall, the manuscript is well organized, it covered most of the studies in the field. But the authors still need to edit the language extensively. Since the reader of a review article may include non-experts, average experts, or new researchers to FMF, a brief and broad introduction, a clear summary of the field and concise conclusions are essential.
e.g.
Line 12: **“The most fearsome complication is secondary amyloidosis due to amyloid A renal deposition in up to 8.6% of cases** or “amyloid A renal deposition induced secondary amyloidosis is the most fearsome complication in up to 8.6% of cases”**
Line 14: **occasionally involve the cardiovascular,**
Line 23: I think that“Beyond” is not appropriate in the context, please consider using “Besides” to replace it.
Line 24: **“such as IL-1 inhibitors anakinra, canakinumab and anti-IL-6 tocilizumab**
etc.
It may help the reader a lot if the authors could simplify the structure of the sentences.
As a review article, it would be great if it has a section for the future directions/ perspectives.
Author Response
Thank you for your observation. We edited the manuscript and provided more details about FMF in introduction.
Line 12: **“The most fearsome complication is secondary amyloidosis due to amyloid A renal deposition in up to 8.6% of cases** or “amyloid A renal deposition induced secondary amyloidosis is the most fearsome complication in up to 8.6% of cases”**
Line 14: **occasionally involve the cardiovascular,**
Line 23: I think that“Beyond” is not appropriate in the context, please consider using “Besides” to replace it.
Line 24: **“such as IL-1 inhibitors anakinra, canakinumab and anti-IL-6 tocilizumab**
It may help the reader a lot if the authors could simplify the structure of the sentences.
Reply: We amended the lines you indicated and revised English to improve the readability of the paper.
Reviewer 2 Report
General
This is an excellent review of Familial Mediterranean Fever (FMF), detailing not only its relationship with renal injuries, but also the general aspects. It describes the genetic background and mechanism of FMF at first, details characteristics of AA-amyloidosis, a risk factor for Mediterranean fever and its renal damage, and touches on the novel treatments. Some minor revisions are required, so they are listed below.
Detailed
1. Some sentences in the discussion are too long to understand, so please revise them. For example, page 14 lines 291 to 295.
2. For some abbreviations, full spelling seems to be better. For examples, coll to colleagues.
3. Please correct some typographical errors; glomerulonephritis (line 103), have to has (line 131), and (line 178).
Author Response
- Some sentences in the discussion are too long to understand, so please revise them. For example, page 14 lines 291 to 295.
Reply: Thank you for your comments. We edited all the paper to simplify the sentences and to improve the comprehension of our text.
For some abbreviations, full spelling seems to be better. For examples, coll to colleagues.
Reply: We amended the abbreviations according to your suggestion.
3. Please correct some typographical errors; glomerulonephritis (line 103), have to has (line 131), and (line 178).
Reply: Thank you for your observations, we corrected all the typographical errors in the paper.
Best regards,